# Field-Layer Vegetation and Water Table Level as a Proxy of CO$_2$ Exchange in the West Siberian Boreal Bog

Danil V. Ilyasov [1,*] , Anastasia V. Meshcheryakova [1], Mikhail V. Glagolev [2,3], Iuliia V. Kupriianova [1], Alexandr A. Kaverin [1], Alexandr F. Sabrekov [1], Mikhail F. Kulyabin [2] and Elena D. Lapshina [2]

[1] Laboratory of Ecosystem Geoinformatics, Yugra State University, 628012 Khanty-Mansiysk, Russia
[2] Laboratory of Ecosystem-Atmosphere Interactions of the Mire-Forest Landscapes, Yugra State University, 628012 Khanty-Mansiysk, Russia
[3] Faculty of Soil Science, Lomonosov Moscow State University, 119991 Moscow, Russia
[*] Correspondence: d_ilyasov@ugrasu.ru

**Abstract:** The Mukhrino field station has participated in the national project on the inventory of carbon fluxes and pools in the terrestrial ecosystems of Russia since 2022. The development of a network of measurements of CO$_2$ fluxes and phytomass covered six types of bog ecosystems typical to Western Siberia. The gross ecosystem exchange (GEE) of the field-layer vegetation (medians for the period from the end of May to the end of July, mgC m$^{-2}$ h$^{-1}$; see errors in Results section) decreased in series: *Sphagnum* bog with sparse low pine trees ("Open bog"), ridges in ridge-hollow patterned bogs ("Ridge"), pine-dwarf shrub-*Sphagnum* bog ("Tall ryam"), hollows in patterned bogs ("S.hollow", "E.hollow") and pine-dwarf shrub-*Sphagnum* bog ("Ryam"): −220, −200, −125, −120, −109 and −86, respectively. Ecosystem respiration (R$_{eco}$) here was 106, 106, 182, 55, 97 and 136. The aboveground and belowground phytomass of mosses in this series varied between $368 \pm 106$–$472 \pm 184$ and $2484 \pm 517$–$6041 \pm 2079$ g/m$^2$, respectively: the aboveground phytomass of vascular plants and plant litter—$15 \pm 7$–$128 \pm 95$ and $10 \pm 6$–$128 \pm 43$, respectively. According to the results of mathematical modeling, the best proxy for GEE, in addition to photosynthetically active radiation and soil surface temperature, was the aboveground phytomass of vascular plants (Ph$_V$), and for R$_{eco}$—Ph$_V$ and the mass of the plant litter of vascular plants.

**Keywords:** west Siberia; Russian carbon inventory project; plant phytomass fractioning; gross ecosystem exchange; ecosystem respiration; mathematical modelling; climate change



## 1. Introduction

Carbon dioxide is the most important greenhouse gas, and its increasing concentration in the atmosphere is the main cause of global warming [1]. Pristine wetland ecosystems play a critical role in the carbon cycle by acting as a natural sink for CO$_2$ and are the largest reservoir of carbon (C). Occupying 3% of the land surface, wetland ecosystems store more carbon than any other terrestrial ecosystems, including forests [2–4]. One of the most waterlogged regions of the world is the West Siberian Lowland (WSL), where wetland ecosystems cover 592,440 km$^2$ and store more than 3% (70 PgC) of the total carbon of terrestrial ecosystems [5–7].

The main mechanism of carbon accumulation in wetland ecosystems is the assimilation of atmospheric CO$_2$ during the photosynthesis of green plants and the subsequent accumulation of carbon in the form of peat deposits (gross-ecosystem exchange—GEE). An opposing process that occurs in wetland ecosystems is the soil and plant respiration, accompanied by oxidation of peat carbon and CO$_2$ emissions into the atmosphere (Ecosystem respiration—R$_{eco}$). The sum of these processes is known as the net ecosystem exchange of carbon dioxide between the ecosystem and the atmosphere (NEE) [6,8–11]. The NEE

of swamps (as well as its components—$R_{soil}$, $R_{eco}$ and GEE) can change under the influence of many interrelated factors: hydrological, climatic, phytocenotic, anthropogenic and others [6,12–15].

Studying the drivers of the magnitude and variability of the carbon dioxide balance (NEE) of wetland ecosystems is an important aspect of furthering scientific knowledge about the functioning of the climate system, inventory of terrestrial sources and sinks of carbon dioxide. It can also help us to find ways to adapt to climate change. However, the current estimates of the carbon balance of terrestrial ecosystems in Russia are extremely uncertain, and their variability is more than 500% [16–27].

Russia's ratification of the Paris climate agreement implies the fulfillment of obligations to limit carbon emissions and the need to form scientifically based national reporting of the carbon balance. In 2022, an innovative project of national importance, the "Unified National Monitoring System for Climatic Substances", was launched in Russia and approved by order of the government of the Russian Federation. As part of the implementation of this project, the creation of a unified national system for monitoring carbon pools and fluxes is envisaged by a consortium of observation stations, one of which is the Mukhrino field station (Mukhrino FS, MFS). The relevance of such a project is extremely high, and the involvement of the MFS will compensate for the lack of data in the future, at least partially, from long-term observations of carbon pools and fluxes in Eurasia, and, in particular, WSL [6,25–27].

In addition to the MFS, there are only a few sites available for the long-term monitoring of greenhouse gases in Western Siberia. Among them is the ZOTTO high-rise tower, maintained by the eddy covariance network along the Yenisei River [28] and the Japan–Russia Siberian Tall Tower Inland Observation Network located in the taiga, steppe and wetland biomes of Siberia [29]. In the southern taiga of Western Siberia, automated chamber observations of methane and carbon dioxide fluxes are being carried out [30], as well as studies related to the carbon balance of peatlands, considering the dynamics of phytomass [6,31].

Mukhrino FS was founded in 2009, and since then, it has accumulated a significant amount of data on the impact of climate change on wetland ecosystems, carbon cycle and biodiversity [32]. The net productivity of vegetation, determined from the growth rate of *Sphagnum*, was used to estimate the rate of carbon input into the ecosystem during the snowless period. The rate of decomposition of plant residues characterizes the relative activity of the microbial transformation of peat under various environmental conditions. Eddy covariance techniques have been used to summarize ecosystem-scale $CO_2$ fluxes. In addition, studies of NEE fluxes of carbon dioxide on a local scale have been carried out by an automatic chamber method in the ridge–hollow complex, considering spatiotemporal variability [6].

However, a deeper understanding of the functioning of wetland ecosystems as components of the carbon dioxide cycle requires considering the components of NEE ($R_{eco}$ and GEE), assessing the magnitude and variability of fluxes, linking the components of the carbon dioxide balance with ecosystem drivers [33] and extending such observations to all the presented types of wetland ecosystems, considering their spatial and ecological diversity. The further accumulation of data obtained using the chamber method will allow them to be mutually verified with the results of measurements by the eddy covariance method and express the patterns obtained by parametrizing mathematical models [34–37]. In addition, the parametrization of mathematical models (considering relationship between carbon dioxide fluxes with abiotic environmental drivers and phytomass) is important for further spatio-temporal upscaling of local data.

Thus, the topic of our work was the fractionation of carbon dioxide fluxes (to NEE, GEE and $R_{eco}$) using the static chamber method and the search for their relationship with the phytomass of vegetation cover and abiotic ecosystem drivers using the example of typical bog ecosystems of the middle taiga of Western Siberia of Mukhrino FS.

## 2. Materials and Methods

### 2.1. Study Location

The Mukhrino Field Station (MFS) is located in the central part of West Siberia in the middle taiga bioclimatic zone (boreal climate), 20 km south-west of Khanty-Mansiysk (N 60.892135, E 68.682330), on the second terrace of the left bank of the Irtysh River (near the confluence with Ob River). The MFS research area is in the northeast part of the Mukhrino pristine mire complex, which covers a total area of ~75 km² (Figure 1). The extensive area to the southwest is represented by the peatland and lake landscapes of the Kondinskaya lowland, interspersed with forests along the rivers. The MFS area covers ~1 km² and has a 2-km-long system of boardwalks, an energy supply complex (solar panels and wind generator) and permanent hydrometeorological and biodiversity monitoring plots for different elements of peatland ecosystems (microtopes). The mean annual amount of precipitation (for the period from 1991 to 2021) and mean annual air temperature are calculated based on the data from three weather stations ("Khanty-Mansiysk", "Ivdel" and "Ugut") and were 549 mm and −0.3 °C, respectively.

The mean air temperature from 1991 to 2021 was −20 °C in winter (January-March, November-December) and 20 °C in summer (May-September). The annual precipitation varied greatly in individual years during three periods: (February–April) from 24 mm to 33 mm, (May–September) from 53 mm to 82 mm, (October–December) from 61 mm to 35 mm from the maximum to the minimum values, respectively.

The Mukhrino mire complex, according to [38], is an oligotrophic raised *Sphagnum* bog. It occupies a local watershed between two small streams, the Mukhrina and Bolshaya rechka, and water discharges to both (see Figure 1). On the eastern side, the Mukhrino bog margin is formed by a terrace scarp that rises 2–6 m above the Mukhrina stream valley. The shape of the scarp is undulated due to active backward erosion by several source brooks of the Mukhrina stream [6].

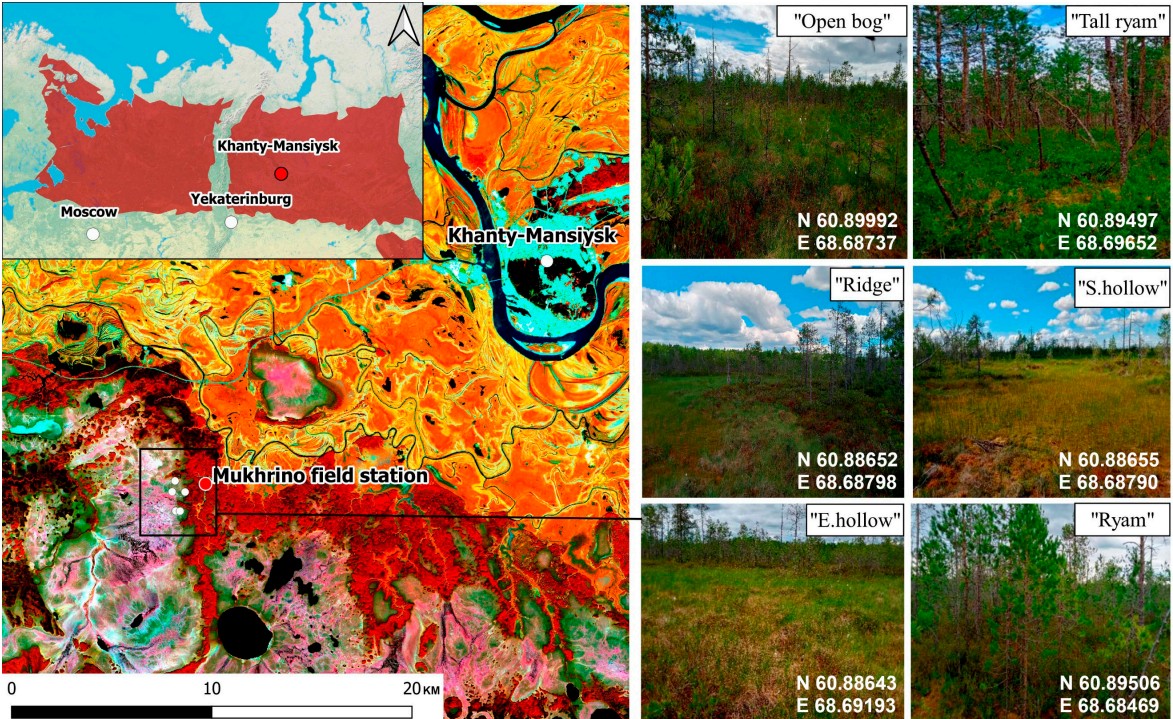

**Figure 1.** Location of research objects: the inset shows the boreal zone [39] on the left and pictures of the studied biotopes on the right.

### 2.2. Vegetation Cover of Study Sites

The objects of this study were bog biotopes typical of the middle taiga of Western Siberia (the number is followed by the name of the site, then the description of the biotope) (Figure 1):

1.  "Open bog": *Sphagnum* bog with sparse low pine trees with *Pinus sylvestris* L., *Chamaedaphne calyculata* (L.) Moench, *Eriophorum vaginatum* L., *Sphagnum angusti-folium* (C.E.O.Jensen ex Russow) C.E.O.Jensen and *S. divinum* Flatberg and K. Hassel. The dwarf pine layer is very sparse or absent. These mire types occur on the border between the oligotrophic raised bog and the mineral uplands. This transition zone usually has a width of 100 to 200 m and is rarely wider. They may also develop in the transition (boundary) zone between the raised bogs and minerotrophic fens.
2.  "Tall ryam": wooded pine-dwarf shrub-*Sphagnum* bog with *P. sylvestris* (6–10 m tall), *Ledum palustre* L., *C. calyculata*, *Vaccinium myrtillus* L., *S. angustifolium* and *S. divinum*; *Sphagnum* species are dominant in the ground layer.
3.  Ridge–hollow patterned bogs. This mire type is the most widespread ombrotrophic patterned bog complex in West Siberia and consists of pine dwarf shrubs-Sphagnum (ryam) ridges and *Sphagnum* hollows, more or less oriented across a rainwater flow. These complexes are usually situated on very slight sloping areas (gradient 0.003–0.008 m/km). The configuration and spacing of the ridges and hollows are related to the slope gradient of the peatland surface, but mostly, they have an equal share in the complex. The ridge microtopes are dryer and 25–50 cm higher than the hollows. We investigated three microtopes:

    3a.  Ridge: Sphagnum ridges with *P. sylvestris*, *L. palustris*, *C. calyculata* and *Sphagnum fuscum* (Schimp.) H.Klinggr. The pine height is usually 0.5–2.0 m with 3–10% cover.
    3b.  S.hollow: hollows occupied by *Scheuchzeria palustris* L.
    3c.  E.hollow: hollows occupied by *E. vaginatum*.
4.  "Ryam": typical ryam with *P. sylvestris* (0.5–4 m tall), *L. palustris*, *C. calyculata* and *S. fuscum*. Ombrotrophic dwarf shrubs: *Sphagnum* hummock peatlands wooded by pine trees. This type of bog is very common in West Siberia and covers large homogeneous areas or is presented as ridges in patterned bog complexes.
5.  Detailed geobotanical description of studied areas (plant heights, the determination of the projective cover, phenophase and vitality) could be found in [40].

### 2.3. Field Data Collection and Analysis

2.3.1. Chamber Measurements of Carbon Dioxide Fluxes

The carbon dioxide fluxes were measured using the static chamber method across six typical bog sites ("Open bog", "Tall ryam", "Ridge", "S.hollow", "E.hollow" and "Ryam"—further in the text, we will use the designations of the sites without quotes) in the late spring and summer of 2022: 31 May–1 June; 24–26 June; and 27 June–28 July [6,41] using an infrared gas analyzer Li-850 (Li-COR Biogeosciences, Lincoln, NE, USA). The measurements were carried out in the daytime, from 10 am to 4 pm local time. A cube chamber made of transparent Plexiglas with the dimensions of $0.4 \times 0.4 \times 0.4$ m and a built-in fan on the upper side of the chamber was used for air mixing. The chamber was installed in a stainless-steel base trough ($0.4 \times 0.4$ m in size) filled with settled water (to ensure the tightness of the internal space of the chamber during measurements), which was cut into the soil to a depth of 10–15 cm. In each of the six biotopes, two chambers (one down and one up) were used to capture the natural spatial variability of the ecosystems. The measurements were performed in three-time repetitions (between measurements, the chamber was ventilated for at least a minute), and the exposure time within each repetition was 2–5 min.

Net ecosystem exchange (NEE) was measured using the transparent chambers, which characterized the net exchange of $CO_2$ between the soil, grass–moss–dwarf shrub layer (in

the presence of shrubs) and the atmosphere; opaque (covered by foiled foam propylene)—$CO_2$ emission into the atmosphere due to the heterotrophic respiration of the soil and the autotrophic respiration of the grass–moss–shrub layer ($R_{eco}$). Gross ecosystem exchange (GEE) characterizing the absorption of $CO_2$ from the atmosphere by the grass–moss (-shrub) layer was calculated as the difference between NEE and $R_{eco}$. Fluxes from the ecosystem to the atmosphere are taken positive, and from the atmosphere to the ecosystem—negative (therefore $R_{eco} > 0$, GEE < 0) [42].

The calculation of specific $CO_2$ fluxes was carried out in the Matlab R2016b software package (MathWorks, Inc., Natick, MA, USA) using the following formula:

$$flux = 2aPMb\frac{H}{(T + T_0)}$$

The absolute error of the calculated fluxes was estimated using the following formula:

$$\Delta flux = 2aPM\frac{1}{(T + T_0)}\left(\Delta bH + \left|b\right|\Delta H\right)$$

where *flux* is the flux of carbon dioxide, mgC m$^{-2}$ h$^{-1}$, *a* is 0.12 mg × mol × K/(kg × J × ppm), *P* is the total pressure of the gas mixture (Pa), *M* is the molar mass of the gas (kg/mol: taken as 0.012 to express the flux in mgC m$^{-2}$ h$^{-1}$), *b* is the rate of change in the gas concentration in the atmosphere of the chamber (ppm/h: calculated as the slope of the direct increase in concentration in the chamber based on the method of least squares, the function "fitlm", MatLab), *H* is the chamber height (m), *T* is the temperature in the chamber at the end of the measurement (K), $T_0$ is the temperature in the chamber at the beginning of the measurement (K), $\Delta b$ is the error in determining parameter *b* (ppm/h) and $\Delta H$ is the error in determining the height of the camera (m: taken as equal to 0.05 m, which is includes half the division value of the measuring tape and the error in determining the height of the chamber, due to the unevenness of the underlying surface).

### 2.3.2. Environmental Conditions

Simultaneously or immediately after the measurements of the carbon dioxide fluxes, the environmental factors were recorded: the water table level (WTL, cm) with a centimeter tape in small pits; electrical conductivity (EC, mS/cm); acidity (pH); and soil temperature (at a depth of 0 ($t_0$) and 15 cm ($t_{15}$), °C) using a Hanna 98130 Combo field multimeter (Hanna Instruments, USA, Woonsocket, RI, USA). The air temperature was recorded at a height of 2 m using temperature loggers: Thermochron iButton DS 1921 (Dallas Semiconductor, Dallas, TX, USA); photosynthetically active radiation (PAR, μmol m$^{-2}$s$^{-1}$) was recorded using a PAR quantum sensor: LI-190R (Li-COR Biogeosciences, Lincoln, NE, USA).

### 2.3.3. Sampling of Vegetation

The sampling of the phytomass of the grass–moss (-shrub, if available) layer was carried out in each biotope after measuring the carbon dioxide fluxes directly at the site of the chambers (in two-fold spatial repetitions) across 31 May–1 June and 24–26 June.

The aboveground phytomass was collected by mowing an area of 40 × 40 cm. The belowground phytomass was collected by cutting peat monoliths 10 × 10 cm in size at depths of 5–15 and 15–25 cm. Under laboratory conditions, the selected samples were divided into the following fractions (according to the methodology that was used in earlier geobotanical studies [43–45]):

(1)  Aboveground phytomass: vascular plants ($Ph_V$, divided by species), green parts of mosses ($Ph_M$, divided into green mosses and *Sphagnum* mosses), the litter of vascular plants ($Ph_L$);

(2)  Belowground phytomass: live tow (photosynthetically inactive parts) of *Sphagnum* and green mosses from a depth of 0–5 cm, living roots and shoots ($R_{5,15,25}$), dead roots and shoots ($MR_{5,15,25}$) from a depth of 0–5, 5–15 and 15–25 cm, other mortmass

and undecomposed peat ($M_{15,25}$) from a depth of 5–15 and 15–25 cm. The selected fractions of the aboveground and belowground phytomass were dried to obtain a constant weight in a laboratory oven at a temperature of 75–80 °C.

### 2.4. Mathematical Modeling

Several simple regression models were used to evaluate the relationship between the net ecosystem exchange of carbon dioxide components (*GEE* and $R_{eco}$) and environmental factors (*WTL*, $T_0$, PAR and plant phytomass). The selected models described the influence of individual environmental factors or the combined impact of their complex [46–48]. The following *GEE* models have been considered:

$$GEE = \left( c_1 Ph_V{}^2 + c_2 Ph_V + c_3 \right) a \frac{\text{PAR}}{k_1 + \text{PAR}} \tag{1}$$

$$GEE = c_4 a Ph_V \frac{\text{PAR}}{k_2 + \text{PAR}} \tag{2}$$

$$GEE = GEE_{\max} \frac{\text{PAR}}{k_3 + \text{PAR}} + c_5 a Ph_V \tag{3}$$

$$GEE = GEE_{\max} \frac{\text{PAR}}{k_4 + \text{PAR}} (1 - \exp(c_6 Ph_V)) \tag{4}$$

$$GEE = GEE_{\max} \frac{\text{PAR}}{k_5 + \text{PAR}} (1 - \exp(c_7 Ph_V)) \exp\left( d_1 WTL^2 + d_2 WTL \right) \tag{5}$$

where *GEE*: gross ecosystem exchange (mgC m$^{-2}$ h$^{-1}$); $c_1$ (m$^2$ g$^{-1}$ h$^{-1}$), $c_3$ (g m$^{-2}$ h$^{-1}$), $c_2$, $c_4$, $c_5$ (h$^{-1}$), $c_6$, $c_7$ (m$^2$/g): parameters of the relationship between GEE and phytomass $Ph_V$; $a$: scaling factor equal 450 (mgC/g); $k_1 \div k_5$: half-saturation constant (such as the value of PAR at which GEE reaches a value equal to half of the maximum); $d_1$ (cm$^{-2}$) and $d_2$ (cm$^{-1}$): GEE and WTL coupling parameters; PAR: photosynthetically active radiation (μmol m$^{-2}$ s$^{-1}$), $Ph_V$: specific aboveground phytomass of vascular plants (g/m$^2$); and $GEE_{\max}$: maximum potential rate of gross photosynthesis (mgC m$^{-2}$ h$^{-1}$), *WTL*: water table level (cm; negative values correspond to WTL above the soil surface, positive values correspond to the below).

Several simple regression models were also used to assess the relationship between the combined respiration of the soil and grass–moss (–shrub) layer and environmental factors:

$$R_{\text{eco}} = Ph_V R_{10} Q_{10}^{\frac{T - T_{ref}}{10}} + d_3 ab WTL \tag{6}$$

$$R_{\text{eco}} = R_{10}' \exp\left( E_0 \left( \frac{1}{T_{ref} - T_0} - \frac{1}{T - T_0} \right) \right) + c_8 a Ph_V \tag{7}$$

$$R_{\text{eco}} = R_{10}'' \exp\left( E_0' \left( \frac{1}{T_{ref} - T_0} - \frac{1}{T - T_0} \right) \right) + c_9 a Ph_L \tag{8}$$

$$R_{\text{eco}} = R_{max}(d_4 WTL + d_5) \tag{9}$$

$$R_{\text{eco}} = R_{10}''' \exp\left( E_0'' \left( \frac{1}{T_{ref} - T_0} - \frac{1}{T - T_0} \right) \right) (1 + d_6 WTL) \tag{10}$$

$$R_{\text{eco}} = (d_7 + d_8 WTL) \exp\left( E_0''' \left( \frac{1}{T_{ref} - T_0} - \frac{1}{T - T_0} \right) \right) + c_{10} a Ph_L \tag{11}$$

where $R_{\text{eco}}$: respiration of the soil and grass–moss (-shrub) layer (mgC m$^{-2}$ h$^{-1}$); $c_8 \div c_{10}$ (h$^{-1}$): parameters of the connection between $R_{eco}$ and specific phytomass $Ph_V$ and $Ph_L$; $a$: scaling factor equal 450 (mgC/g); $b$: scaling factor equal 0.01 (m/cm); $d_3$ (g m$^{-3}$ h$^{-1}$), $d_4$, $d_6$, $d_8$ (cm$^{-1}$), $d_5$, $d_7$ (dimensionless): parameters of the connection between $R_{eco}$ and

*WTL*; $Ph_V$: specific aboveground phytomass of vascular plants (g/m$^2$); $Ph_L$: specific mass of vascular plant litter (g/m$^2$); $R_{10}$: specific respiration rate at 283.85 K and WTL = 0 (mgC g$^{-1}$ h$^{-1}$); $R_{10'}$: respiration rate at 283.85 K and $Ph_V$ = 0 (mgC m$^{-2}$ h$^{-1}$); $R_{10'}'$: respiration rate at 283.85 K and $Ph_L$ = 0 (mgC m$^{-2}$ h$^{-1}$); $R_{10'}''$: respiration rate at 283.85 K and WTL = 0 (mgC m$^{-2}$ h$^{-1}$); $Q_{10}$: dimensionless Van't Hoff temperature coefficient; *T*: soil temperature (K); $T_{ref}$: reference soil temperature (K; taken equal 283.15 K); $T_0$: the temperature at which respiration rate reaches zero (K; taken equal 227.13 K [47]; when $Ph_V$ and $Ph_L$ reaches 0 for models 7 and 8, 11, respectively); *WTL*: water table level (cm); $E_0$, $E_{0'}$, $E_{0'}'$, $E_{0'}''$: activation energy divided by the gas constant (K); and $R_{max}$ set equal 592.2 mgC m$^{-2}$ h$^{-1}$. Identification of the model parameters was carried out by minimizing the sum of the absolute values of the deviations of the model predictions from the "corridor" of the experimental data, specified by the average values ± measurement error. If the deviation from the mean was within this "corridor" (i.e., its absolute value did not exceed the error), then it was considered zero and did not contribute to the objective function. Root mean square error (RMSE) was also carried out as a measure of model fitting. Modeling was performed based on all field data obtained without separation by biotope types.

### 2.5. Data Analysis

The descriptive statistics of the field measurements (min, max, means, standard deviations, medians and IQR) were provided for the measured fluxes. In the case of regression analysis, the threshold value of *p* was set to 0.05. Statistic differences was checked using the Kruskal–Wallis test (for each of 6 sites, threshold value *p* = 0.05; with recourse to nonparametric statistics since the values were not normally distributed), for pairwise comparison.

## 3. Results

### 3.1. Water Table Level and Soil Temperature

The average (±std) water table level (WTL) for the entire measurement period increased in the series Ryam—Tall ryam—Ridge—Open bog—E.hollow—S.hollow and amounted to 45 ± 24—43 ± 25—33± 10—22 ± 15—14 ± 17—6 ± 7 cm below the soil surface. In all of the studied plots (except Ridge), the highest WTL was observed on 31 May–1 June (spring field campaign); by 24–26 June, WTL dropped by 5–20 cm, and by 27–28 July, it reached its minimum for three field campaigns, falling by another 2–40 cm (Figure 2).

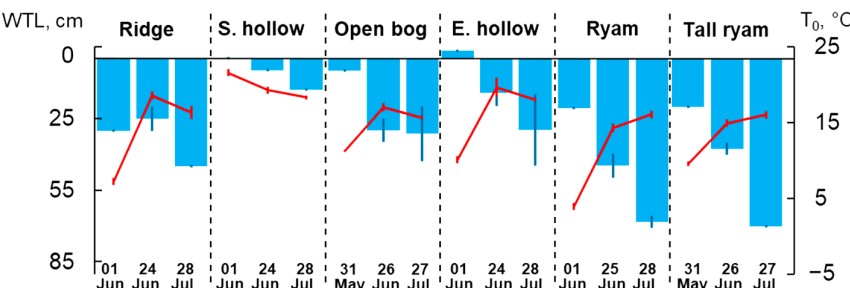

**Figure 2.** Spatial means of water table level, cm (blue columns; bars—min and max values) and soil surface temperature, °C (red lines; bars—min and max values) at investigated sites during measurements of carbon dioxide fluxes.

A slightly different WTL dynamic was at the Ridge, where the WTL was similar at the first and second field campaigns (FC) and at the Open bog, where it was identical in the second and third FC. The highest WTL amplitude for the considered months (40–50 cm) is typical for the Tall ryam and Ryam located on the outskirts of the bog area and the most drained ones; a smaller amplitude of WTL fluctuations was noted for Open bog, E.hollow

and Ridge (15–20 cm), and the minimum was noted on the most humid S. hollow (up to 10 cm).

The average soil surface temperature ($T_0$) differed insignificantly among the considered plots, from $14 \pm 6$ (Ridge) to $20 \pm 2$ °C (S.hollow). The dynamics of $T_0$ in all plots (except for S.hollow) were identical: a minimum (up to $4 \pm 1$ °C in Ryam) on 31 May–1 June, a maximum on 24–26 June (up to $20 \pm 1$ °C in E.hollow) and some decline by 27–28 July. On S.hollow, the maximum soil surface temperature ($22 \pm 1$ °C) was observed in spring, with a gradual decrease towards June and July to $18 \pm 1$ °C. The average soil temperature at a depth of 15 cm ($T_{15}$) varied within close limits, from $16 \pm 5$ (Ridge) to $20 \pm 6$ °C (S.hollow). The pH of the bogs varied from 3.2 in Tall ryam to 4.1 in S.hollow; EC from 30 in E.Hollow to 100 µS/cm in Ryam. Thus, within the framework of our observations, we were able to cover a wide range of the hydrological and temperature soil conditions typical of these habitats.

### 3.2. Carbon Dioxide Fluxes

The median values of NEE (mgC m$^{-2}$ h$^{-1}$; IQR) for the period from May to July increased in the series (Figure 3) Ridge—S.hollow—Open bog—E.hollow—Ryam—Tall ryam and amounted to $-54$ (78), $-47$ (58), $-42$ (61), $-7$ (83), 50 (211) and 158 (280).

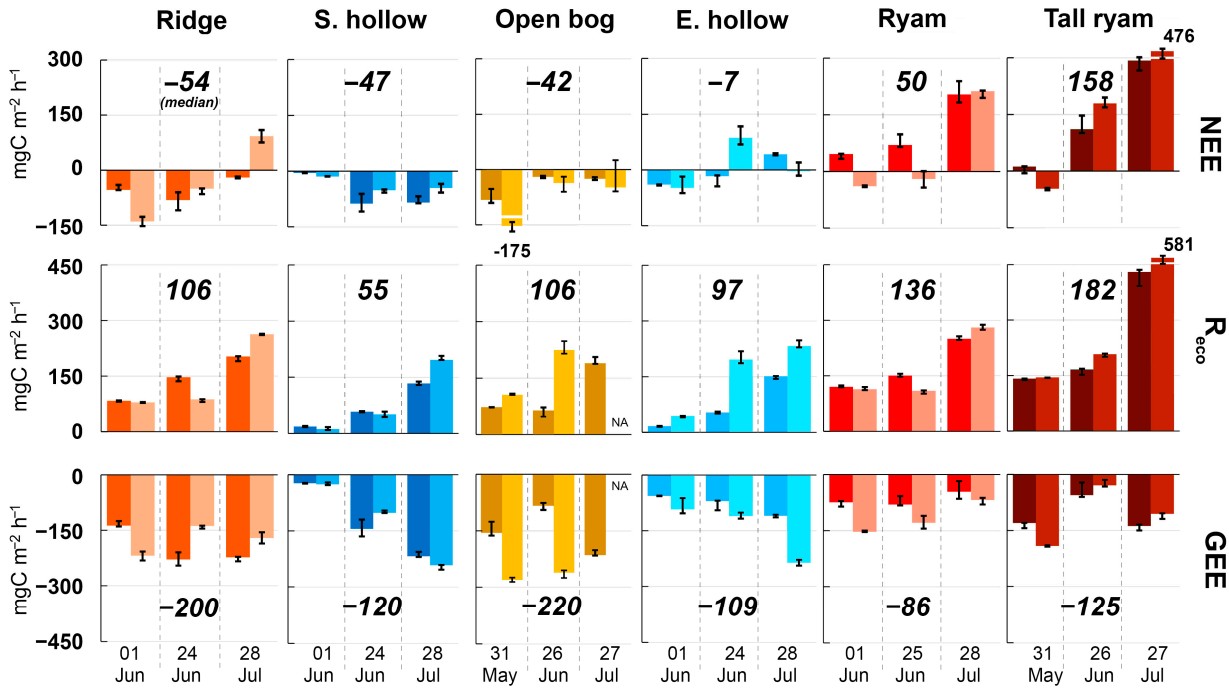

**Figure 3.** Fluxes of $CO_2$ (mgC m$^{-2}$ h $^{-1}$) in May, June, and July in the considered types of wetland ecosystems. Light columns are depressions, dark columns are elevations (spatial repetitions within the biotope); error—the first and third quartiles, the values in the center are the median fluxes for the entire period.

Thus, the plants of the lower tier in Ridge, S.hollow and Open bog are close in value ($-54 \ldots -42$ mgC m$^{-2}$ h$^{-1}$) as absorbers of carbon dioxide; E.hollow has a $CO_2$ balance hovering around zero, with grass–moss–dwarf shrub layer (in the presence of shrubs) in Ryam and Tall ryam as its sources (on average, from May to July, taking into account the spatial variability).

The lowest median $R_{eco}$ was obtained with moistened S.hollow 55 mgC m$^{-2}$ h$^{-1}$ (112), somewhat larger (and close to each other) on the more drained Open bog, Ridge and E.hollow: 106 (124), 106 (115) and 97 (147), respectively, and the largest was obtained from the most drained Ryam and Tall ryam: 136 (134) and 182 (268).

Calculated as the difference between the NEE and $R_{eco}$ medians, the GEE values were the largest in modulus at the Open bog and Ridge sites: $-220$ (115) and $-200$ (86); somewhat smaller on S.hollow, E.hollow and Tall ryam: $-120$ (191), $-109$ (82) and $-125$ (90), respectively; finally, the lowest GEE medians in modulus were obtained from Ryam: $-86$ (79) mgC m$^{-2}$ h$^{-1}$.

In summary, several more aspects can be noted: NEE increased from May to July at all plots (with the exception of S.hollow), and the same dynamics were typical for $R_{eco}$. GEE increased from May to July in hollows with Scheuchzeria palustris and Eriophorum vaginatum; in Ryam, it decreased during this period, and there was no trend in the other areas.

### 3.3. Phytomass

The temporal and spatial heterogeneity of mean ($\pm$std) phytomass of vascular plants and mosses of the grass– moss–shrub layer is shown on Figure 4 and Table 1. Aboveground phytomass of vascular plants (Ph$_V$) was characterized by significant variation from site to site, which reached an order of magnitude between hollows and the rest of the sites. The aboveground moss phytomass for the same series was significantly higher and differed little between sites and field companies (in contrast to the Ph$_V$), with the exception of Ridge and Ryam sites in the end of June (Figure 4). Thus, the main source of variability in aboveground phytomass values from May to the end of June was the spatial one of Ph$_V$.

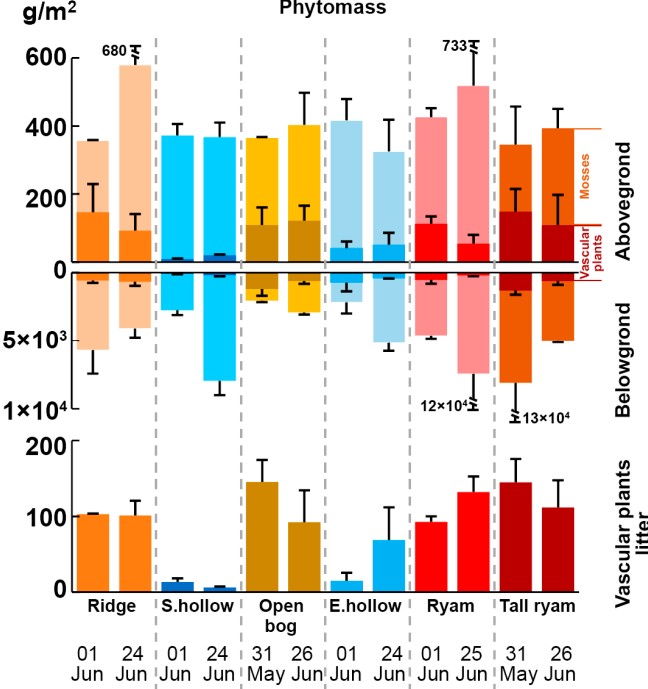

**Figure 4.** Temporal heterogeneity of aboveground phytomass of vascular plants (dark at the top), mosses (light at the top), belowground phytomass of vascular plants (total at a depth of 0–25 cm; dark in the middle), lint, and belowground phytomass of mosses (total at a depth of 0–25 cm; light middle) and vascular plant litter (below) in late spring and early summer. The error is the maximum observed values (symmetrically with the minimum ones).

Spatiotemporal mean ($\pm$std) stocks of belowground phytomass of vascular plants (0–25 cm) were highest on Open bog (908 $\pm$ 551) and Tall Ryam (965 $\pm$ 518) sites, and lowest on S.hollow site; thus, the spatiotemporal variability of below-ground biomass has also reached almost one order of magnitude. The belowground moss phytomass varied significantly less ($\times$2.5 times) and were highest on Ryam (6041 $\pm$ 2079) site, and lowest on Open bog site (2484 $\pm$ 517). The belowground phytomass of both vascular plants and

mosses varied more strongly than the aboveground one. Finally, the mean (±std) in the time and space litter weight of the vascular plants were lowest and similar in hollows (10–42) and highest and similar in all other sites (102–128). It is natural that the mass of vascular plant litter varied in the same way as the mass of living vascular plants.

**Table 1.** Spatial heterogeneity and fractioning of phytomass reserves in various components of grass-moss (-shrub) layer.

| | Phytomass, g/m$^2$ ± std/% of Total | | | | | |
|---|---|---|---|---|---|---|
| | **Aboveground (Vascular Plants)** | **Aboveground (Mosses)** | **Belowground (Vascular Plants)** | **Belowground (Mosses)** | **Litter (Vascular Plants)** | **Total** |
| Ridge | 120 ± 84/2 | 468 ± 152/8 | 625 ± 282/10 | 4910 ± 1773/79 | 102 ± 16/2 | 6225 |
| S.hollow | 15 ± 7/<1 | 370 ± 44/7 | 136 ± 94/3 | 4684 ± 2272/90 | 10 ± 6/0 | 5215 |
| Open bog | 115 ± 56/3 | 384 ± 80/10 | 908 ± 551/23 | 2484 ± 517/62 | 119 ± 52/3 | 4010 |
| E.hollow | 46 ± 33/1 | 371 ± 105/9 | 578 ± 546/14 | 3049 ± 1949/75 | 42 ± 48/1 | 4086 |
| Ryam | 84 ± 43/1 | 472 ± 184/7 | 371 ± 300/5 | 6041 ± 2079/85 | 113 ± 29/2 | 7081 |
| Tall ryam | 128 ± 95/2 | 368 ± 106/5 | 965 ± 518/13 | 5960 ± 5108/79 | 128 ± 43/2 | 7549 |
| Mean | 85/1 | 406/7 | 597/10 | 4521/79 | 86/2 | 5694 |

Ryam and Tall ryam are characterized by the highest total phytomass (total aboveground and belowground phytomass of vascular plants, litter and mosses: 10,018 ± 3824 and 10,581 ± 2779, respectively), while the Open bog is characterized by the lowest (4493 ± 3828). The largest contribution to the stock of both aboveground (from 5% in Tall ryam to 10% in Open bog) and belowground (from 62% in Open bog to 90% in S.hollow) phytomass in all communities is made by mosses. In ecosystems with a characteristic low water table level (Ryam and Tall ryam), the belowground organs of the plants, grasses and shrubs make a significant contribution to the total phytomass, and due to the developed microrelief, these values are characterized by the greatest spatial and temporal variability. The coefficient of variation in the plot-average values of the aboveground phytomass for vascular plants was 54%, for mosses, 12%; belowground, 53 and 33%, respectively; and vascular plant litter, 56%.

*3.4. Link between Carbon Dioxide Fluxes and Phytomass*

The correlation matrix of the median carbon dioxide fluxes and phytomass stock components in the studied swamp areas was first built in order to determine those phytomass components that are most closely related to carbon dioxide fluxes ($R_{eco}$ and GEE) in the context of the parameterization of mathematical models (Table A1).

A close relationship (R = −0.7, *p* = 0.05) was found between NEE and mortmass at a depth of 15 cm ($M_{15}$). In addition, a fairly high correlation coefficient (R = 0.6) was found between the mass of the living roots of vascular plants at the same depth ($R_{15}$) and NEE. Other phytomass components showed a weak negative relationship with carbon dioxide fluxes (|R| did not exceed 0.5). There was no linear relationship between net ecosystem exchange (NEE), vascular plant litter mass ($Ph_L$), dead rootstock at 15–25 cm ($RM_{15,25}$) and other mortmass at 25 cm ($M_{25}$) (i.e., |R| did not exceed 0.1).

$R_{eco}$ directly correlated with vascular plant litter weight ($Ph_L$; R = 0.9); feedback was found with aboveground moss phytomass ($Ph_M$; R = −0.7). The amount of $Ph_L$ differed significantly between different sites; therefore, the high correlation coefficient probably illustrates not only the significant contribution of litter decomposition to $R_{eco}$ but also the rather large spatial variability of $Ph_L$. On the other hand, the negative relationship between $Ph_M$ and $R_{eco}$ probably indicates the effect of moisture conditions on respiration intensity: with an increase in WTL and, as a result, aboveground moss phytomass, respiration processes slow down due to the formation of anaerobic soil conditions. A fairly close relationship (|R| = 0.6) was found between $R_{eco}$ and the aboveground phytomass of vascular plants ($Ph_V$), live roots at a depth of 5 cm ($R_5$) and dead roots at 5 and 15 cm ($RM_{5,15}$), as well as other mortmass at 25 cm ($M_{25}$). At the same time, there is an inverse relationship between dead phytomass and a direct relationship with living phytomass.

Finally, GEE correlated closely with $|R| \geq 0.7$ with aboveground vascular plant phytomass ($Ph_V$), litter ($Ph_L$), and live plant root mass at 15 cm depth ($R_{15}$). $R_{eco}$ and GEE were also closely correlated. As GEE increased, $R_{eco}$ decreased, and vice versa. We assume that the correlation of GEE with $Ph_V$ is directly related to the mass of photosynthetic organs of vascular plants, and GEE with $Ph_L$ is indirectly related: the greater the mass of photosynthetic organs of vascular plants, the greater the mass of litter they form. This assumption is confirmed by the significant correlation between $Ph_V$ and $Ph_L$ ($R = 0.8$). In addition, other plant phytomass components were directly correlated with $Ph_V$: litter ($Ph_L$), live roots at a depth of 5 and 15 cm ($R_{5,15}$), and dead roots at a depth of 5 cm ($RM_5$); at the same time, an inverse correlation was observed with all components of the mortmass.

Based on the results obtained, the following phytomass fractions were selected when parameterizing the relationship between carbon dioxide fluxes and phytomass components: for GEE, the aboveground phytomass of vascular plants ($Ph_V$), and for $R_{eco}$, the mass of vascular plant litter ($Ph_L$) and $Ph_V$.

### 3.5. Model Parametrization

To parameterize the models, in addition to $Ph_V$ and $Ph_L$, the following were used as independent variables: photosynthetically active radiation (PAR) and (or) WTL for GEE; surface soil temperature ($T_0$) and/or WTL for $R_{eco}$. Figures 5 and 6 show the results of model parametrization comparing to the results of the field measurements (see parameters of models in Table A2). The correlation coefficient shown in the figure illustrates the relationship between the modeled and measured fluxes as a whole for the model and not as a single independent variable.

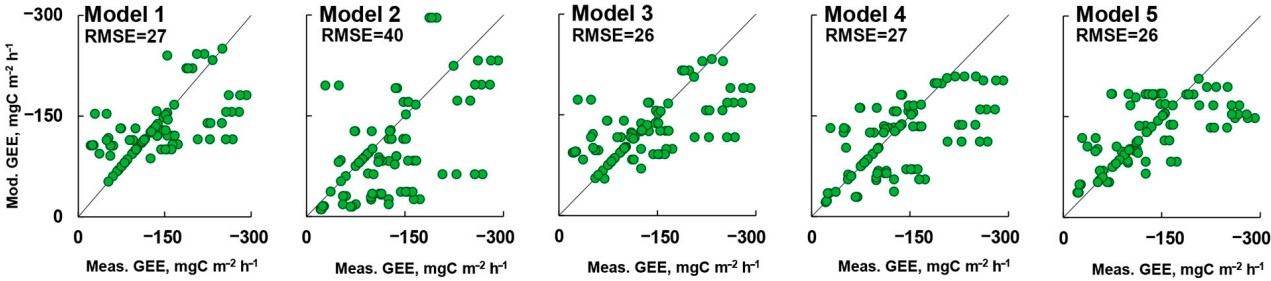

**Figure 5.** Relationship between modelled (Y-axis; using different models—see Section 2.4) and observed (X-axis) gross ecosystem exchange (GEE).

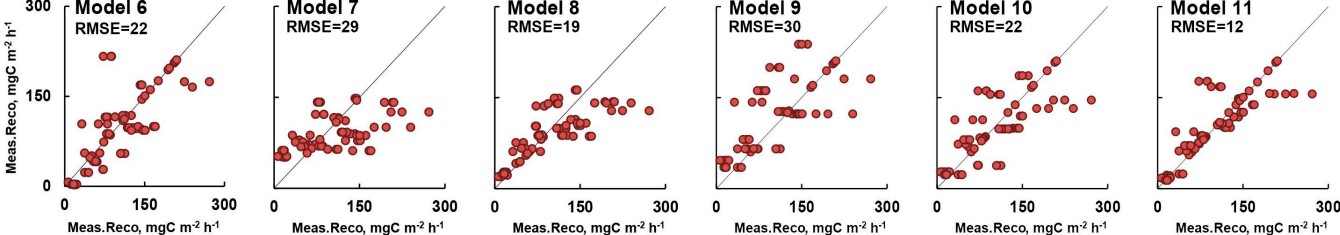

**Figure 6.** Relationship between modelled (Y-axis; using different models—see Section 2.4) and observed (X-axis) ecosystem respiration ($R_{eco}$).

The correlation coefficient (R) between the simulated and measured GEE fluxes decreased in the model 4, 5, 3, 2 and 1 and amounted to 0.68, 0.66, 0.63, 0.62 and 0.55; RMSE was similar for models 4, 5, 3 and 1 and amounted to 27, 26, 26 and 27, respectively, but 40 for model 2. The GEE–PAR relationship is characterized by an accumulation curve; the half-saturation constants $k_1 \div k_5$ were 202, 248, 487, 330 and 275 μmol m$^{-2}$s$^{-1}$ for models 1–5, respectively. The GEE–$Ph_V$ relationship in models 4 and 5 is also represented by an accumulation curve (the GEE ceases to change by more than 5% that is achieved at $Ph_V = 125$ and 75 g/m$^2$, respectively); in other cases, it is continues to grow linearly (model

2, 3) or accelerating (model 1) to phytomass reserves. Finally, the relationship between GEE and WTL is shown in model 5, which has a bell-shaped curve. Unfortunately, considering the GEE–WTL connection did not significantly improve the simulation results: models 2, 3 and 4 showed the similar R when describing the field measurement data.

As a result of the parameterization of the $R_{eco}$ models, the correlation coefficient (R) decreased in the series model 11, 9, 6, 8, 10, 7 and amounted to 0.83, 0.82, 0.77, 0.76, 0.74, 0.55, respectively; RMSE was 12, 30, 22, 19, 22, 29, respectively (Figure 6).

The temperature dependence in the models is described by a power-law (model 6) or exponential function (models 7, 8, 10 and 11): the coefficient $Q_{10}$ for model 6 was 4.9, and in other cases, the respiration rate increased by 1.2–1.9 with an increase in temperature by 10 °C for the range of 10–20 °C and 1.2–2.0 times for the range of 18–28 °C. In contrast to the GEE, the account in the $R_{eco}$ models of the connection between the processes of respiration and the WTL played a more important role: it was the models that took into account WTL variability in one way or another that were better parameterized. In addition, accounting for the litter mass of vascular plants played an important role in the $R_{eco}$ variability. In models 7 and 8, which are identical in structure, the best parameterization result was achieved in the latter, including the dependence on $Ph_L$ (Table A2). However, a significant improvement in the results of parameterization of model 8 in comparison with model 7 can be explained by the fact that the amount of litter was correlated with WTL (R = 0.61).

In summary, it can be noted that the models that consider the relationship with 1–2 independent variables (models 2 and 9) showed an R that is not inferior to the more complex models (models 5 and 11). Successful parameterization of the GEE relationship model with PAR and aboveground phytomass will allow in the future, as data are accumulated, to disseminate the results of modeling in time and space; however, this will require considering the daily dynamics of PAR and the spatiotemporal variability of phytomass stocks. The $R_{eco}$ models could be parameterized more precisely because ecosystem drivers (WTL and phytomass) vary less over fluxes measurements than PAR. However, the temperature dependence of $R_{eco}$ looks relatively ambiguous, which will require closer research. In addition, an important task of improving the predictive capacity of models can be the parametrization of models individually for each of the considered ecosystems.

## 4. Discussion

### 4.1. Carbon Dioxide Fluxes and Phytomass

According to the literature data, the ecosystem respiration ($R_{eco}$) in the swamp ecosystems of the boreal zone with similar ecological conditions varies from 38 in the *Sphagnum* bog [47] up to 132 ± 49 in shrub communities of ombrotrophic bogs [49,50] and 194 ± 96 mgC m$^{-2}$ h$^{-1}$ in the ryams [38,51]. The measurements made on the bog massif studied by us earlier in the ridge–hollow complex showed $R_{eco}$ varying from 56 to 132 in ridges and from 28 to 75 mgC m$^{-2}$ h$^{-1}$ in hollows [6]. The minimum $R_{eco}$ values obtained because of the current study were observed in the S.hollow (hollow with the dominance of *Scheuchzeria palustris* L.) in early June and amounted to 11 (median; 9-1Q, 17-3Q), and the maximum values were observed in the Tall ryam at the end of July and reached 581 (572, 587) mgC m$^{-2}$ h$^{-1}$. At the same time, the median $R_{eco}$ in these areas from the end of May to the end of July was 55 and 182 mgC m$^{-2}$ h$^{-1}$, respectively (Table 2). In general, our results are consistent with the available estimates: the $R_{eco}$ values obtained by us in Tall ryam may seem somewhat overestimated. These measurements were carried out in areas where the stocks of aboveground phytomass of vascular plants (148 ± 67 g/m$^2$) and litter (145 ± 31) did not practically differ from the literature data (128 ± 42 and 189 ± 114 g/m$^2$, respectively [31,52–54]). It is likely that such $R_{eco}$ values at the Tall ryam, Ridge and S.hollow sites may be due to the coincidence of two other factors favoring an increase in the intensity of respiration processes: (1) a significant decrease in WTL by the end of 2 July (2) the maximum temperature of the surface of the peat layer, which was observed at the Tall ryam during the same period (Figure 2). Given the relatively small temporal variability of phytomass stocks from late May to late June, we suggest that

WTL and soil temperature were the most important drivers of $R_{eco}$ flux variability not only in space (both between and within different sites) but also in time, as it was found out in [46–48].

The intensity of carbon dioxide assimilation, GEE (from $-86$ in Ryam to $-200$–$-220$ mgC $m^{-2}\,h^{-1}$ in Ridge and Open bog), seems to be somewhat higher in comparison to the literature data. For example, on similar ridges in an oligotrophic bog in European Russia, GEE varied between $-185$–$-84$ mgC $m^{-2}\,h^{-1}$, in hollows and pools here $-150$–$-46$ mgC $m^{-2}h^{-1}$ [50,55,56]; and on the current object of study according to earlier measurements, [6] $-157$–$-50$ mgC $m^{-2}$ $h^{-1}$. In an oligotrophic bog in the middle taiga [9], the rate of carbon assimilation did not exceed $-50$ mgC $m^{-2}\,h^{-1}$, in an ombrotrophic bog in Finland around $-80$–$-106$ mgC $m^{-2}\,h^{-1}$ [57]. Thus, our results are closer to the upper limit (the highest absorption of carbon dioxide) of the GEE estimates according to the literature data. We assume that the reason for this is the method of organizing the measurements that we used: all of the measurements were taken during the period when the sun was high above the horizon (from 10 am to 4 pm local time), and mostly in cloudless weather. Accounting for diurnal GEE variability and taking measurements under more varied lighting conditions would not only provide a more adequate average GEE estimate but also better parameterize the GEE–PAR relationship in Models 1–5.

**Table 2.** Net ecosystem exchange, ecosystem respiration, and gross ecosystem exchange in bog ecosystems according to the literature data.

| Zone/Ecosystem | Site Description | CO$_2$ flux, mgC m$^{-2}$ h$^{-1}$ | | Component of CO$_2$ Balance | Note | Reference |
|---|---|---|---|---|---|---|
| South taiga (European Russia)/oligotrophic bog | Site 1 [1] | 118 105–155 | | $R_{eco}$ | mean min–max | [56] |
| | Site 2 [2] | 170 164–176 | | | | |
| | Ridges | 151 138–184 | | | | |
| | Hollows | 92 84–98 | | | | |
| | Ridges and hummocks | 131–149/$-185$–$-84$ | | $R_{eco}$/GEE | min–max | [55] |
| | Hollows Swamps | 51–99 | /$-150$–$-85$ /$-115$–$-46$ | | | |
| Middle taiga (West Siberia)/oligotrophic bog | Ridge Hollow | $-21$–11 $-39$–$-11$ | | NEE | min–max | [58] |
| South taiga (West Siberia)/oligotrophic bog | Ryam | $78 \pm 12$—$194 \pm 96$ | | $R_{eco}$ | mean $\pm$ std | [51] |
| Middle taiga (West Siberia)/forest | Pine forest | 84/$-120$/$-35$ | | $R_{eco}$/GEE/NEE | mean | [9] |
| Middle taiga (West Siberia)/oligotrophic bog | Sphagnum bog | 38/$-50$/$-7$ | | | | |
| Steppe (East Siberia)/steppe | Feather grass steppe | 82/$-108$/$-26$ | | | | |
| South tundra (East Siberia)/tundra | Southern and tussock tundra | 42/$-44$/$-2$ | | | | |

**Table 2.** *Cont.*

| Zone/Ecosystem | Site Description | $CO_2$ flux, mgC m$^{-2}$ h$^{-1}$ | Component of $CO_2$ Balance | Note | Reference |
|---|---|---|---|---|---|
| Temperate forest (Poland)/mesotrophic peatland | Site 1 [3] | $161 \pm 127$–$376 \pm 107/-219 \pm 156$ | $R_{eco}$/NEE | mean ± std | [59] |
| | Site 2 [4] | $126 \pm 19$–$237 \pm 85/-142 \pm 108$ | | | |
| | Site 3 [5] | $127 \pm 59$–$187 \pm 57/-178 \pm 94$ | | | |
| | Site 4 [6] | $136 \pm 44$–$267 \pm 95/-219 \pm 122$ | | | |
| Middle taiga (West Siberia): oligotrophic bog | Ridge-hollow complex | $-49/-100/-106/-22$ | NEE | May/ … /August | [60] |
| South taiga (West Siberia): mesotrophic peatland | Poor fen | $-249$–$32$ | NEE | May–October (2013–2019) | [61] |
| South taiga (West Siberia): oligotrophic bog | Ridge Hollow | 12–177 43–116 | $R_{eco}$ | Min–max (August) | [62] |
| Mesotrophic peatland | Poor fen | 20–204 | | (July) | |
| Middle taiga (West Siberia): oligotrophic bog | Ryam | 1–7 | $R_{eco}$ | Grass-moss layer | [63] |
| Sub-Taiga (West Siberia): fens Forest-steppe (West Siberia): fens | Reed fen Reed fen Willow-hummock-sedge fen | 114–241128–3372–662 | $R_{eco}$ | min-max | [64] |
| Middle taiga (West Siberia)/oligotrophic bog | Ridge Hollow | $56$–$132/-157$–$-75$ $28$–$75/-99$–$-50$ | $R_{eco}$/GEE | mean | [6] |
| Boreal climate (Western Finland)/ombrotrophic bog | Hummocks | $63$–$68/-69$–$-80$ | $R_{eco}$/GEE | mean | [10] |
| Cold temperate humid climate (Northern Sweden)/oligotrophic bog | Carpets and lawns complex | $6.4$–$4.4/-11.4$–$-11.1/-5.1$–$-6.7$ | $R_{eco}$/GPP/NEE | mean | [11] |
| Southern boreal zone (Southern Finland)/ombrotrophic bog | Hollow, lawn and hummock complex | $-21$–$-113$ | GPP | min-max | [65] |

Note: [1] Pinus sylvestris, Carex pauciflora, *Sphagnum* spp.; [2] Pinus sylvestris, Vaccinium microcarpum, V. uliginosum, *Sphagnum* spp.; [3] Caricetum elatae; [4] Calamagrostietum ignoreae; [5] Menyantho-Sphagnetum teretis; [6] Sphagno apiculati-Caricetum rostratae: all units are recalculated in mgC m$^{-2}$ h$^{-1}$.

Finally, the median net ecosystem exchange (NEE) varied from uptake at a rate of $-54$ mgC m$^{-2}$ h$^{-1}$ at the Ridge site to emission at a rate of 158 mgC m$^{-2}$ h$^{-1}$ at the Tall ryam site, depending on which of the main components of the dioxide balance carbon (GEE or $R_{eco}$) was found to be prevalent. Earlier studies [58] showed a near-zero balance on ridges ($-21$–$11$) and absorption in hollows ($-39$–$-11$) based on the results of measurements during the growing seasons within several years of observations. According to our results, NEE in Ridge and S.hollow was biased more towards assimilation. We assume that in addition to the possible overestimation of GEE, as mentioned above, this may be because the measurements were carried out up to the peak of vegetation of plants (end of July) and during the rest of the summer in such ecosystems, a shift in the carbon dioxide balance towards $CO_2$ losses to the atmosphere [6,58]. Bog ecosystems, along with forests, tundra and steppes, according to estimates by various authors [6,9,55,59–61], function as a sink of carbon dioxide with varying intensities: from $-125$ mgC m$^{-2}$ h$^{-1}$ in temperate Sphagnum bog [66] and $-106$ in the oligotrophic bogs of the middle taiga to $-219 \pm 122$

in mesotrophic mires of the temperate forest zone and $-249$ in south taiga. At the same time, the $R_{eco}$ in the swamps of other climatic zones [61–64] or other mineral nutrition varies in an extremely wide range from 12 to 337 mgC m$^{-2}$ h$^{-1}$. It should be emphasized that our measurements were made in field-layer vegetation, and future consideration of tree vegetation will certainly change the resulting estimates for Ridge, Open bog, Ryam, and Tall Ryam; however, the carbon dioxide balance of individual canopy components is important in terms of their contribution to the overall $CO_2$ cycle [67].

As noted above, the results of estimating the aboveground phytomass stocks of vascular plants are in good agreement with the data obtained in similar ecosystems of Western Siberia [52,53]. The mass of the green (photosynthetic) parts of mosses varied in our plots from $325 \pm 93$–$391 \pm 59$ in E.hollow and Tall ryam to $579 \pm 101$–$519 \pm 214$ g/m$^2$ in Ridge and Ryam, respectively. According to [52], in hollows, ryam, and tall ryam, the moss phytomass reached $369 \pm 75$, $370 \pm 88$ and $285 \pm 78$ g/m$^2$, which generally agrees with our results. In [53], for swamps of Western and Eastern Siberia in various natural zones, the green mass of mosses varied from 39 (167)–210 g/m$^2$ in ridges to 37–450 g/m$^2$ in hollows. It is possible that the moss phytomass obtained in our study looks somewhat overestimated in Ridge and Ryam, and it would be natural to trace its effect on carbon dioxide fluxes, but such a dependence could not be found. In general, considering the spatial variability and the influence of humid conditions in different years on the amount of moss growth [53], our results are in good agreement with the literature. The same can be noted with regard to the value of the belowground plant phytomass: in [52,54], the weight of roots of grasses and shrubs in tall ryam, ryam and hollow was estimated at $669 \pm 276$, $556 \pm 122$ and $584 \pm 313$ g/m$^2$, and in similar ecosystems in the framework of this work in $620 \pm 273$–$1310 \pm 300$, $205 \pm 55$–$537 \pm 277$ and $420 \pm 10$–$736 \pm 631$, which, taking into account spatial variability (lower values of phytomass were obtained in depressions, large in elevations), is almost identical. The mass of moss lint in the same ecosystems in [52] varied between $895 \pm 639$–$4075 \pm 724$, and in our work, $1010$–$2480 \pm 1540$ g/m$^2$; vascular plant litter $164 \pm 75$–$232 \pm 101$ and $145 \pm 31$–$69 \pm 43$ g/m$^2$, respectively. The last two fractions of the vegetation cover in our work, on the contrary, look underestimated but are close in value to [52]. It is likely that the accumulation of a larger array of factual data will make it possible to obtain a better agreement between the data (this is especially important in the context of the subjectivity of the methodology used: the green parts of mosses and the division of roots into living and dead ones largely depend on the experience and methodological approach used by a researcher). Our results also are very close to those obtained in [68] which were also carried out on oligotrophic bogs in the Khanty-Mansiysk Autonomous Okrug. This also applies to the proportion of phytomass (except for the tree layer): in [68], in ridges and ryams increases the proportion of phytomass of vascular plants and decreases the proportion of mosses, and vice versa in hollows. Furthermore, moss phytomass of pristine peatlands in Finland was about 700 g m$^{-2}$, that a little bigger obtained in our study [69]; vascular plant phytomass here was much bigger—700 g m$^{-2}$ [69] which, as we assume, is associated with milder climatic conditions and a longer vegetation period of the plant.

The results obtained from the assessment of carbon dioxide fluxes, phytomass reserves and the relationship of these components will be supplemented in the future (it is necessary to assess the contribution of woody vegetation, increase the number of sampling sites, measurements of fluxes and phytomass stocks) and used to calculate the carbon dioxide balance of the entire bog massif, taking into account the spatial-temporal variability of ecosystem drivers.

### 4.2. Modelling

In order to evaluate the modeling results of individual components ($R_{eco}$ and GEE) and the total carbon dioxide balance (NEE), we summed the $R_{eco}$ and GEE obtained from all the models used in a pairwise fashion and compared the obtained results with the measured NEE (Table 3). In addition to the results obtained based on individual models, the median fluxes of $R_{eco}$ and GEE were calculated (column and row "med", Table 3).

**Table 3.** Correlation (R) between measured and simulated NEE values. The simulated NEE values are the pairwise sum of the results of the GEE and $R_{eco}$ simulations based on models 1–5 and 6–11, respectively, and the median flow values obtained from these models (see Section 2.4).

| | | GEE Model | | | | | |
|---|---|---|---|---|---|---|---|
| | | **1** | **2** | **3** | **4** | **5** | **Med** |
| | 6 | 0.6 | 0.6 | 0.6 | 0.6 | 0.7 | 0.7 |
| | 7 | 0.6 | 0.4 | 0.6 | 0.4 | 0.4 | 0.6 |
| | 8 | 0.5 | 0.4 | 0.5 | 0.5 | 0.6 | 0.5 |
| $R_{eco}$ model | 9 | 0.6 | 0.5 | 0.5 | 0.5 | 0.6 | 0.6 |
| | 10 | 0.6 | 0.5 | 0.6 | 0.6 | 0.7 | 0.6 |
| | 11 | 0.6 | 0.5 | 0.6 | 0.6 | 0.6 | 0.6 |
| | med | 0.6 | 0.5 | 0.6 | 0.6 | 0.6 | 0.6 |

Note: bright green shows |R| ≥ 0.8.

The best fit (R = 0.7, $p < 0.05$) between the modeled and measured NEEs was achieved using pairs of models 5–6, 5–10, med–6 (median based on GEE models 1–5). Thus, the best result in NEE modeling was achieved using with model pair with model 5: it is noteworthy that this is the only one of the GEE models, that includes dependency from WTL. It also considers the value of phytomass not as a simple proportional term and the response of plants to PAR growth in the form with saturation. A similar relationship was used in model 4, which gave an even better result when modeling GEE only (however, this did not improve the predictive power for NEE). In addition, as noted above, considering that WTL improve the predictive power for GEE modelling, it was, however, important in the calculation of $R_{eco}$: the sum of GEE with $R_{eco}$ models 6 and 10 yielded the best NEE simulation result. The most complex $R_{eco}$ model that consider the largest number of various parameters in a non-linear way (models 11) as well as simplest one (model 9) show similar and successful results: it is possible that on the one hand, complication of models does not always lead to better results, but on the other hand, additional information on the receipt of new fluxes and associated environmental drivers will allow to be better parameterized complex model in the future. It is noteworthy that an equally good result was obtained when summing model 5 with models 6 and 10, although the last one does not consider aboveground phytomass of vascular plants ($Ph_V$). It is very likely that the aboveground phytomass of vascular plants is a good indicator of long-term moisture conditions, and these values are correlated with WTL. Therefore, when modeling carbon dioxide fluxes within individual ecosystems, it is necessary to consider ecosystem drivers more accurately, which will vary within a smaller range [47]. This will require more long-term and extensive fieldwork.

**5. Conclusions**

The data from the long-term net ecosystem exchange (NEE) measurements taken at Mukhrino FS earlier are supplemented and fractionated using the static chamber method into constituent components (GEE and $R_{eco}$) in six different bog biotopes, considering the spatial-temporal variability. The lower tier of vegetation in Ridge, S.hollow and Open bog accumulate carbon dioxide; E.hollow has a $CO_2$ balance hovering around zero, Ryam and Tall ryam acts as its sources. The largest contribution to the stock of both aboveground and belowground phytomass in all communities is made by mosses. As a result of the parameterization of several simple regression models, it was found that, in addition to abiotic environmental factors (PAR for GEE; WTL and $T_0$ for $R_{eco}$), the most important proxy for carbon dioxide fluxes is the aboveground vascular plant phytomass and vascular plant litter. In contrast, the moss cover did not show a relationship with the components of the carbon dioxide balance and flux variability, despite being the largest phytomass fraction. The parameterization of the models over the entire data set (with simultaneous consideration of all types of ecosystems) first describes the spatial variability of $CO_2$ fluxes depending on the considered ecosystem drivers. At the same time, simpler regression mod-

els (as well as medians of model values) showed a similar results of modelling relationship between carbon dioxide fluxes and independent variables as well as complex ones.

**Author Contributions:** Conceptualization, D.V.I.; methodology, D.V.I. and A.V.M., software, D.V.I.; investigation, D.V.I., I.V.K. and A.A.K.; writing—original draft preparation, D.V.I., A.V.M., A.F.S. and M.V.G.; writing—review and editing, I.V.K., M.V.G., E.D.L., M.F.K. and A.F.S.; visualization, D.V.I. and A.V.M.; supervision, I.V.K., M.V.G., E.D.L., M.F.K. and A.F.S. All authors have read and agreed to the published version of the manuscript.

**Funding:** The research was supported by the state assignment of Ministry of Science and Higher Education of the Russian Federation to organize a new young researcher Laboratory in Yugra State University (FENG-2022-0001) as a part of the implementation of the National Project "Science and Universities" (supplementary agreement No. 075-03-2022-169/5).

**Institutional Review Board Statement:** Not applicable.

**Informed Consent Statement:** Not applicable.

**Data Availability Statement:** Data is contained within the article.

**Acknowledgments:** We kindly appreciate Aleksey Dmitrichenko and all engineers and technical staff who provided logistics on boats to Mukhrino FS, Egor Dyukarev for support with LI-850 and providing information about the development of Mukhrino FS, Artur Niyazov for logistics support, Arina Bikulova for field measurements and Tatyana Zavitaeva for phytomass processing.

**Conflicts of Interest:** The authors declare no conflict of interest. The funders had no role in the design of the study; in the collection, analyses, or interpretation of data; in the writing of the manuscript, or in the decision to publish the results.

## Appendix A

**Table A1.** Correlation matrix of carbon dioxide fluxes and phytomass components in the five studied types of wetland biotopes.

| $Ph_V$ | $Ph_M$ | $Ph_L$ | $R_5$ | $R_{15}$ | $R_{25}$ | $RM_5$ | $RM_{15}$ | $RM_{25}$ | $M_{15}$ | $M_{25}$ | NEE | $R_{eco}$ | GEE | |
|---|---|---|---|---|---|---|---|---|---|---|---|---|---|---|
| 1.0 | −0.3 | 0.8 | 0.7 | 0.8 | 0.0 | 0.8 | −0.5 | −0.4 | −0.3 | −0.8 | −0.3 | 0.6 | −0.7 | $Ph_V$ |
| | 1.0 | −0.5 | −0.6 | −0.3 | 0.4 | −0.6 | 0.0 | 0.0 | 0.4 | 0.1 | −0.5 | −0.7 | 0.2 | $Ph_M$ |
| | | 1.0 | 0.7 | 0.5 | 0.3 | 0.7 | −0.6 | −0.4 | −0.5 | −0.7 | 0.1 | 0.9 | −0.7 | $Ph_L$ |
| | | | 1.0 | 0.2 | −0.3 | 0.7 | −0.4 | −0.1 | −0.9 | −0.5 | 0.5 | 0.6 | −0.2 | $R_5$ |
| | | | | 1.0 | 0.2 | 0.9 | 0.0 | −0.1 | 0.2 | −0.4 | −0.6 | 0.4 | −0.8 | $R_{15}$ |
| | | | | | 1.0 | 0.0 | 0.0 | 0.1 | 0.1 | 0.0 | −0.5 | 0.2 | −0.5 | $R_{25}$ |
| | | | | | | 1.0 | 0.0 | 0.1 | −0.3 | −0.4 | −0.2 | 0.6 | −0.6 | $RM_5$ |
| | | | | | | | 1.0 | 0.9 | 0.3 | 0.9 | −0.1 | −0.6 | 0.4 | $RM_{15}$ |
| | | | | | | | | 1.0 | −0.1 | 0.9 | 0.1 | −0.4 | 0.4 | $RM_{25}$ |
| | | | | | | | | | 1.0 | 0.1 | −0.7 | −0.4 | −0.2 | $M_{15}$ |
| | | | | | | | | | | 1.0 | 0.1 | −0.6 | 0.6 | $M_{25}$ |
| | | | | | | | | | | | 1.0 | 0.2 | 0.6 | NEE |
| | | | | | | | | | | | | 1.0 | −0.7 | $R_{eco}$ |
| | | | | | | | | | | | | | 1.0 | GEE |

Note: bright green shows $|R| \geq 0.8$, light green $|R| = 0.7$; $Ph_V$: aboveground phytomass of vascular plants (field-layer vegetation); $Ph_M$: aboveground phytomass of mosses; $Ph_L$: litter of vascular plants; $R_{5,15,25}$: live belowground phytomass of vascular plants from a depth of 5, 15 and 25 cm; $RM_{5,15,25}$: dead belowground phytomass of vascular plants from a depth of 5, 15 and 25 cm; $M_{15,25}$: other mortmass from a depth of 15 and 25 cm; NEE: net ecosystem exchange; $R_{eco}$: ecosystem respiration; GEE: gross ecosystem exchange.

**Table A2.** Parameter values obtained during the parametrization of mathematical models.

| Model | Parameter | | | | |
|---|---|---|---|---|---|
| 1 | $c_1$<br>$-5.2 \times 10^{-6}$ | $c_2$<br>$-3.5 \times 10^{-4}$ | $c_3$<br>$-0.3$ | $k_1$<br>$202.5$ | |
| 2 | $c_4$<br>$-3.8 \times 10^{-3}$ | $k_2$<br>$248.3$ | | | |
| 3 | $c_5$<br>$-1.4 \times 10^{-3}$ | $k_3$<br>$486.5$ | $GEE_{max}$<br>$-123.7$ | | |
| 4 | $c_6$<br>$-1.7 \times 10^{-2}$ | $k_4$<br>$329.8$ | $GEE_{max}$<br>$-271.0$ | | |
| 5 | $c_7$<br>$-4.7 \times 10^{-2}$ | $k_5$<br>$274.6$ | $d_1$<br>$-6.3 \times 10^{-4}$ | $d_2$<br>$3.4 \times 10^{-2}$ | $GEE_{max}$<br>$-152.7$ |
| 6 | $d_3$<br>$0.6$ | $R_{10}$<br>$1.7 \times 10^{-1}$ | $Q_{10}$<br>$4.9$ | | |
| 7 | $c_8$<br>$9.7 \times 10^{-4}$ | $R_{10}$<br>$40.3$ | $E_0$<br>$132.6$ | | |
| 8 | $c_9$<br>$1.8 \times 10^{-3}$ | $R_{10}'$<br>$5.1$ | $E_0'$<br>$693.8$ | | |
| 9 | $d_4$<br>$6.5 \times 10^{-3}$ | $d_5$<br>$7.8 \times 10^{-2}$ | | | |
| 10 | $d_6$<br>$9.9 \times 10^{-2}$ | $R_{10}''$<br>$20.0$ | $E_0''$<br>$237.7$ | | |
| 11 | $c_{10}$<br>$1.1 \times 10^{-2}$ | $d_7$<br>$7.4$ | $d_8$<br>$8.6 \times 10^{-1}$ | $E_0'''$<br>$425.6$ | |

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
