# Peer review of "Field-Layer Vegetation and Water Table Level as a Proxy of CO2 Exchange in the West Siberian Boreal Bog"

_land, doi:10.3390/land12030566_

Round 1

Reviewer 1 Report

The study on ‘Field-Layer Vegetation and Water Table Level as a Proxy of CO2 Exchange in the West Siberian Boreal Bog’ by Ilyasov et al., is a well-structured manuscript and contains useful information that can enhance knowledge of mathematical modeling for the aboveground phytomass of vascular plants (PhV), and for Reco—PhV and the mass of the plant litter of vascular plants. However, in my opinion, the manuscript needs some minor additions/modification to improve the article.

1. Page-1: I find some irreverent citations in the introduction part. For example, [6–11] and [12–17]. Remove some.

2. Fig. 1: Is it necessary to use reference in the fig. legend? Please correct if necessary. Moreover, I suggest to reproduce it using longitude and latitude.  

3. Materials and methods part is too long. Please consider to re-write it.

4. Please consider Table 2 as Table S1.

5. Some sections of discussion part cited poorly. Please improve the discussion part with more recent citations.   

Overall, from my observation and search, it’s a unique study. It can be recommended for publication after addressing the comments above.

Author Response

Dear reviewer!

We are thanking very much for your attention to our manuscript and we greatly appreciate the comments. We have tried to improve the article in accordance with the comments made. Below we respond in detail to the comments:

  1. Page-1: I find some irreverent citations in the introduction part. For example, [6–11] and [12–17]. Remove some.

- Thank you very much for your valuable comment. We have reviewed the articles we cite. And removed those of them that did not relate to the statement in the text (also we change them to more relevant ones – please, see in references section):

Chestnykh, O. V.; Grabovskiy, V. I.; Zamolodchikov, D. G. Estimate of the Soil Carbon Stock of Forested Regions in Russia Using Databases of Soil Properties. Contempor. Probl. Ecol. 2022, 15(7), 731–740. https://doi.org/10.1134/S1995425522070071

Olchev, A.; Novenko, E.; Desherevskaya, O.; Krasnorutskaya, K.; Kurbatova, J. Effects of climatic changes on carbon dioxide and water vapor fluxes in boreal forest ecosystems of European part of Russia. Environ. Res. Lett. 2009, 4, 045007.

Golovatskaya, E.A.; Dyukarev, E.A. The influence of environmental factors on the CO2 emission from the surface of oligotrophic peat soils in West Siberia. Eurasian Soil Sci. 2012, 45, 588–597.

Naumov, A.V. Soil Respiration; Izd SO RAN: Novosibirsk, Russia, 2009; pp. 1–208.

Sasakawa, M.; Ito, A.; Machida, T.; Tsuda, N.; Niwa, Y.; Davydov, D.; Fofonov, A.; Arshinov, M. Annual variation of CH4 emissions from the middle taiga in West Siberian Lowland (2005–2009): A case of high CH4 flux and precipitation rate in the summer of 2007. Tellus B Chem. Phys. Meteorol. 2012, 64, 17514.

Veretennikova, E.E.; Dyukarev, E.A. Diurnal variations in methane emissions from West Siberia peatlands in summer. Russ. Meteorol. Hydrol. 2017, 42, 319–326.

Veretennikova, E.E.; Dyukarev, E.A. Spatial and temporal dynamics of methane fluxes from the bog ecosystems of the southern taiga of Western Siberia. Boreal Environ. Res. 2021, 26, 43–59.

But we did not delete some, because here was considered NEE as a sum of CO2 emission (Respiration) and CO2 assimilation (GEE or GPP):

Dyukarev, E.; Zarov, E.; Alekseychik, P.; Nijp, J.; Filippova, N.; Mammarella, I.; Filippov, I.; Bleuten, W.; Khoroshavin, V.; Ganasevich, G.; et al. The Multiscale Monitoring of Peatland Ecosystem Carbon Cycling in the Middle Taiga Zone of Western Siberia: The Mukhrino Bog Case Study. Land 2021, 10, 824. https:// doi.org/10.3390/land10080824

Bubier, J.L.; Bhatia, G.; Moore, T.R.; Roulet, N.T.; Lafleur, P.M. Spatial and temporal variability in growing-season net ecosystem carbon dioxide exchange at a large peatland in Ontario, Canada. Ecosystems 2003, 6, 353–367.

Helfter, C.; Campbell, C.; Dinsmore, K.J.; Drewer, J.; Coyle, M.; Anderson, M.; Skiba, U.; Nemitz, E.; Sutton, M.A. Drivers of long-term variability in CO2 net ecosystem exchange in a temperate peatland. Biogeosciences 2015, 12, 1799–1811.

Molchanov, A.G.; Olchev, A.V. Model of CO2 exchange in a sphagnum peat bog. Comput. Res. Model. 2016, 8, 369–377.

Leroy, F.; Gogo, S.; Guimbaud, C.; Bernard-Jannin, L.; Hu, Z.; Laggoun-Défarge, F. Vegetation composition controls temper-ature sensitivity of CO2 and CH4 emissions and DOC concentration in peatlands. Soil Biol. Biochem. 2017, 107, 164–167.

  1. 1: Is it necessary to use reference in the fig. legend? Please correct if necessary. Moreover, I suggest to reproduce it using longitude and latitude.

- We would prefer to leave a reference to the boundaries of the boreal zone just below the figure, because they were created based on the map of the cited source. In addition, we have added the geographic coordinates of the plots to the right side of the figure.

  1. Materials and methods part is too long. Please consider to re-write it.

- The Materials and Methods section of our manuscript occupies 11,700 characters with spaces out of a total of 42,300 characters (excluding abstract, bibliography, and other ancillary sections), which is just over 25% of the article. We strive to make the methodology of work as clear as possible to the reader and try to reproduce it as fully as possible. A significant proportion of articles in journals contain a comparable amount of description of the methods used, which is the norm. Moreover, we do not go beyond the permissible limits of the scope of the publication.

  1. Please consider Table 2 as Table S1.

- Thank you very much for this comment, we have moved the table to the Appendix section.

  1. Some sections of discussion part cited poorly. Please improve the discussion part with more recent citations.

- Thank you very much for this comment, we have add some resent publications.

Thank you for your consideration of this manuscript!

Reviewer 2 Report

Comments to the manuscript by Ilyasov et al.

This manuscript explored the Field-Layer Vegetation and Water Table Level as a Proxy of CO2 Exchange in the West Siberian Boreal Bog.

The manuscript was written according to the results of hard work in the field and lab. It has a good idea to represent the fractionation of carbon dioxide fluxes using the static chamber method and the search for their relationship with the phytomass of vegetation cover and abiotic ecosystem drivers using the example of typical bog ecosystems of the middle taiga of Western Siberia of Mukhrino FS.

In general, I have main observations:

(1) The language is acceptable.

(2) The study locations paragraph has no information about the weather; please, add the average of at least five years of the climate factors of the study site.

(3) The study has a few references in the discussion and materials and methods, so please, add more references. For instance, "Sampling of Vegetation."

(4) In the first mention of the plant species in the text, write the full name with the authors; after that, write the abbreviation such as "Pinus sylvestris L.", abbreviation "P. sylvestris".

(5) Generally, the tables must have a head on every page, such as table 2. 

(6) In Vegetation Cover of Study Sites, plant species need more descriptive information such as family, life form, size (high & width), and others.

(7) "Environmental Conditions" have to be highlighted and linked with other factors in the results and discussion.

Author Response

Dear reviewer!

We are thanking very much for your attention and we greatly appreciate the comments. We have tried to improve the article in accordance with the comments made. Please, see full answer below:

  1. The language is acceptable.

- Thank You!

  1. The study locations paragraph has no information about the weather; please, add the average of at least five years of the climate factors of the study site.

- we calculate meteorological conditions as a mean of 3 weather stations and calculate average for the 30 – years period from 1991 to 2021 as it usually done for “current climate” in meteorology. This section we added to 2.1 “Study location” (last sentence of the first paragraph). Thank you!

  1. The study has a few references in the discussion and materials and methods, so please, add more references. For instance, "Sampling of Vegetation."

- we add more references in these sections.

  1. In the first mention of the plant species in the text, write the full name with the authors; after that, write the abbreviation such as "Pinus sylvestris ", abbreviation "P. sylvestris".

- Full name with the authors updated for abbreviations in section 2.2.

  1. Generally, the tables must have a head on every page, such as table 2.

- Thank you for your valuable comment, we have tried to place the tables in such a way that they are entirely on one page. This was not possible for table 3, so we added a title to the second page. Also, we move table 2 to supplement material (table S1).

  1. In Vegetation Cover of Study Sites, plant species need more descriptive information such as family, life form, size (high & width), and others.

- A detailed geobotanical description was not included in the goals of our study of carbon dioxide fluxes and their relationship with the total phytomass of the vegetation cover. They are described much more detailed in the specialized study to which we referred:

Filippov I. V.; Lapshina E. D. Peatland unit types of lake-bog systems in the Middle Priob’ie (Western Siberia). Environm. Dyn. and Glob. Clim. Change. 2008, 1(1S), 115–124.

  1. "Environmental Conditions" have to be highlighted and linked with other factors in the results and discussion.

- The main goal of our work was to find the relationship between carbon dioxide fluxes and phytomass. This goal was achieved by obtaining model parameters linking phytomass and fluxes, which are presented in Table A1. In order to structure the manuscript, we have singled out separate sections in the results section, demonstrating the consistent expression of the values of fluxes (3.2. Carbon Dioxide Fluxes), phytomass (3.3. Phytomass), their relationships (3.4. Link between Carbon Dioxide Fluxes and Phytomass) and expression in the form of parametrized models (3.5. Model Parametrization). We followed the same sequence of data discussion in the discussion section: 4.1. Carbon Dioxide Fluxes and Phytomass; 4.2. Modelling. In Section 4.1, we do not discuss in detail each component of flux association with phytomass that were presented in Section 3.4 and Table 2, since here our task was to find the most important components, namely PhV and PhL. We have tried to discuss these factors in detail in section 4. In addition, we have added additional citations to this section. We did not discuss the groundwater level and temperature in detail in Section 4, since the relationships of them with carbon dioxide fluxes is widely known in the cited references:

Ilyasov, D.V.; Molchanov, A.G.; Glagolev, M.V.; Suvorov, G.G.; Sirin, A.A. Modelling of carbon dioxide net ecosystem exchange of hayfield on drained peat soil: Land use scenario analysis. Comput. Res. Model. 2020, 12, 1427–1449.

Laine, A.; Riutta, T.; Juutinen, S.; Väliranta, M.; Tuittila, E.S. Acknowledging the spatial heterogeneity in modelling/reconstructing carbon dioxide exchange in a northern aapa mire. Ecol. Model. 2009, 220, 2646–2655.

Riutta, T.; Laine, J.; Tuittila, E.S. Sensitivity of CO2 exchange of fen ecosystem components to water level variation. Ecosystems 2007, 10, 718–733

Thank you for your consideration of this manuscript!

Reviewer 3 Report

The subject of the research is appropriate. The good and valuable traits have been evaluated. The experiments are conducted well. Overall, I see this study an interesting piece of work, but needs substantial revisions. The introduction section should be updated by recent and related references. Please make sure that the research problem is explicitly clear to the reader. The results should be revised and % increase or decrease should be mentioned. In discussion, the comparison/support with the previously published results experiments is not completely compatible. Moreover, it should be updated with recent references. Conclusion could be more concise focusing on the significant concluding statements. Authors need to check grammar, space, comma, mis-spell, large and small letters, and others. Good luck!

Author Response

Dear Reviewer,

Thank you very much for your attention and we greatly appreciate the comments. We have tried to improve the article in accordance with the comments made. 

Based on comments from you and the editor, we have corrected the presentation of the introduction, results, discussion and conclusion sections. Added a new table to simplify the perception of numerical information. We also revised and developed with current ones the references to the literature used in the article. In addition, we tried to correct the overall perception of the manuscript.

Reviewer 4 Report

I find the Manuscript entitled " Field-Layer Vegetation and Water Table Level as a Proxy of CO2 Exchange in the West Siberian Boreal Bog" is   very interesting and a worthwhile contribution to our knowledge. I think that the information in the paper should be published eventually.

Author Response

Dear Reviewer,

Thank you very much for your attention and we greatly appreciate the comments. We have tried to improve the article in accordance with the comments made. 

Based on comments from Reviewer 1 and the editor, we have corrected the presentation of the introduction, results, discussion and conclusion sections. Added a new table to simplify the perception of numerical information. We also revised and developed with current ones the references to the literature used in the article. In addition, we tried to correct the overall perception of the manuscript.